# Diagnosing Failure Root Causes in Platform-Orchestrated Agentic Systems: Dataset, Taxonomy, and Benchmark

## Abstract

Agentic systems consisting of multiple LLM-driven agents coordinating through tools and structured interactions, are increasingly deployed for complex reasoning and problem-solving tasks. At the same time, emerging low-code and template-based agent development platforms (e.g., Dify) enable users to rapidly build and orchestrate agentic systems, which we refer to as **platform-orchestrated agentic systems**. However, these systems are also fragile and it remains unclear how to systematically identify their potential failure root cause. This paper presents a study of root cause identification of these platform-orchestrated agentic systems. To support this initiative, we construct a dataset **AgentFail** containing 307 failure logs from ten agentic systems, each with fine-grained annotations linking failures to their root causes. We additionally utilize counterfactual reasoning-based repair strategy to ensure the reliability of the annotation. Building on the dataset, we develop a taxonomy that characterizes failure root causes and analyze their distribution across different platforms and task domains. Furthermore, we introduce a benchmark that leverages LLMs for automatically identifying root causes, in which we also utilize the proposed taxonomy as guidance for LLMs. Results show that the taxonomy can largely improve the performance, thereby confirming its utility. Nevertheless, the accuracy of root cause identification reaches at most 33.6%, which indicates that this task still remains challenging. In light of these results, we also provide actionable guidelines for building such agentic systems. In summary, this paper provides a reliable dataset of failure root cause for  platform-orchestrated agentic systems, corresponding taxonomy and benchmark, which serves as a foundation for advancing the development of more reliable agentic systems.

## 1 Introduction

Large Language Models (LLMs) have recently shown remarkable capabilities in reasoning, planning, and knowledge-intensive tasks, which promotes their widely adoption across diverse application domains Chen et al. (2023). Building on these advances, agentic systems powered by LLMs are gaining increasing attention, as they enable multiple specialized agents to collaborate toward complex goals. Such agentic systems have been applied to software development, information retrieval and research assistance, where the coordination of agents often outperforms single-agent solutions.

To further lower the barrier to building such systems, a new wave of low-code agentic AI development platforms, such as Dify Dify Contributors (2023), Coze Coze Contributors (2023), n8n n8n Contributors (2020) and AutoGen Wu et al. (2023a), has emerged. These platforms provide intuitive workflow editors, pre-configured tool integrations, and flexible orchestration mechanisms, allowing users to rapidly prototype and deploy multi-agent solutions without extensive programming expertise.

Although these low-code and template-based tools have significantly lowered the bar for building agentic systems, the inherent fragility of agents and the errors introduced by the workflow and platform make these built systems prone to failure in practice. However, pointing out the root cause of failures is challenging because the systems often involve a large number of interconnected nodes with complex dependencies, which makes root cause localization inherently difficult.

Recent research has begun to explore failure attribution in multi-agent systems. Zhang et al. (2025d) proposed Who&When dataset with annotated responsible agent and failure step and benchmarks for automated attribution. Methods like AgenTracer Zhang et al. (2025a) attempted to identify responsible agents or actions using spectrum analysis. ETO Song et al. (2024) improved the performance of LLM agents in complex tasks by enabling them to learn from failed explorations. These efforts highlight the importance of understanding failure mechanisms, but they largely target traditional algorithm-generated (e.g., AG2 Wang et al. (2024)) or hand-crafted agentic systems (e.g., CAMEL Li et al. (2023)), rather than the emerging platform-orchestrated agentic systems which enables the rapid construction and coordination of multiple agents on a platform. In addition, prior work on failure attribution effectively identifies where a failure occurs (e.g., a specific agent). Yet they do not know why it occurs (i.e., the cause of the occurrence), e.g., a poor prompt design or a workflow deadlock. These deeper understanding is the basis for the follow-up failure analysis, repair, and enhancement.

To bridge this gap, we present an empirical study of root cause identification in platform-orchestrated agentic systems, which focus on systematically analyzing why failures arise. Specifically, our contributions are threefold: (i) Dataset[1] - We construct 307 annotated failure data obtained from two representative platforms with multi-round expert annotation and reliable expert consensus. Each data instance contains the query content, the workflow and configuration of the corresponding system, the execution failure logs, and the annotated root cause of failure. Additionally, we draw on the idea of counterfactual reasoning and use targeted repair experiments to ensure the reliability of the annotations. (ii) Failure Root Cause Taxonomy – We construct a fine-grained taxonomy of agent-, workflow-, and platform-level failures, grounded in analysis of failure traces. Furthermore, we quantitatively analyze failure distributions across platforms and tasks, providing comparative insights into failure patterns from different perspectives. (iii) Benchmark - We conduct benchmark experiments on multiple LLMs (e.g, `gpt-4o`) for assessing the ability of automated failure root cause diagnosis, and results show that incorporating our taxonomy into LLM prompts significantly improves their accuracy in identifying root causes. Nevertheless, the maximum accuracy of root cause identification is 33.6%, which implies the challenge of this task. Finally, based on our insights, we provide practical guidelines for developers to construct more robust agentic systems on platform, ensuring that our study not only deepens understanding but also supports real-world platform practices.

## 2 BACKGROUND

### 2.1 AGENTIC AI DEVELOPMENT PLATFORM

Modern agentic AI development platforms (e.g., Dify, Coze, n8n) expose agentic system construction through a node-based paradigm, where each node corresponds to a functional unit. By connecting nodes, users can define complex multi-agent system that combine reasoning, control, and external interactions. The **node types** are summarized as follows, the detailed description of the nodes can be found in the Appendix A.2: 1) Start and Termination Nodes; 2) LLM and Agent Nodes; 3) Knowledge Nodes; 4) Logic and Control Nodes; 5) Code and Template Nodes; 6) Tool and Integration Nodes. These integration nodes extend the platform's capability beyond built-in components, enabling workflows to interact with external services, databases, or custom tools.

Together, these nodes provide support for the following functions: 1) Reasoning and task execution; 2) Knowledge search; 3) Dynamic orchestration; 4) Data transformation and management; 5) External tool integration; This modular design allows practitioners to flexibly compose workflows that combine autonomous reasoning, structured control, and tool use, lowering the barrier to building multi-agent systems while maintaining extensibility.

### 2.2 FAILURE DEFINITION

#### 2.2.1 TASK FAILURE DETERMINATION

Since we would like to collect failure trajectories of agentic systems, we first need to determine whether the current execution has deviated from expected behavior. In our setting, the execution of system interrupts is directly regarded as a failure. For the concluded executions, we adopt the following two evaluation strategies for different types of tasks:

---

[1]Repo URL: https://anonymous.4open.science/r/ICLR26-27B2/

**Ground Truth Comparison.** For tasks with deterministic ground truth or validation mechanism (e.g., code generation), we directly compare the system outputs against the provided ground truth. One execution is considered successful if the output either passes test-based verification (e.g., functional correctness in code execution) or exactly matches the reference solution.

**Multi-LLM Judge.** For tasks lacking deterministic ground truth (e.g., task planning, deep research), we adopt LLM-as-a-judge Gu et al. (2024) technique. Specifically, we use multiple independent LLMs as evaluators, each tasked with assessing whether the system output satisfied the task requirements. Then, a consensus voting strategy is applied, where an output was marked as correct if a majority of LLM judges agreed on its validity. Example prompt we uses is in Appendix A.4.

### 2.2.2 FAILURE LOCALIZATION

Following previous work Zhang et al. (2025d), we denote the outcome of a task trajectory $\tau$ by a binary variable $\phi(\tau)$, where $\phi(\tau) = 1$ indicates failure and $\phi(\tau) = 0$ indicates success. Failure or success of the trajectory is determined in Section 2.2.1.

To capture the failure root cause, we firstly need to find the decisive error made by agents. We suppose that agent $i$ makes a wrong action $a_t$ at step $t$ in a failed trajectory $\tau$. Therefore, the error can be represented as $E(i, t)$. Then, we construct a modified trajectory by replacing $a_t$ with a correct action $\hat{a}_t$ while keeping prior steps unchanged and the following steps are adjusted according to the new correct action. If this intervention changes the trajectory outcome from failure to success, we believe that the error $a_t$ at step $t$ is decisive. Formally, the decisive error indicator is defined as:

$$\Delta_{E(i,t)}(\tau) = \begin{cases} 1, & \text{if } \phi(\tau) = 1 \text{ and } \phi(\tau^{E(i,t)}) = 0, \\ 0, & \text{otherwise.} \end{cases} \quad (1)$$

$\Delta_{E(i,t)}(\tau) = 1$ means fixing agent $i$'s error at time $t$ changes the trajectory from fail to success. Since multiple decisive mistakes may exist, we follow the earliest-in-time principle, i.e., selecting the first decisive error as the root of system failure:

$$E(i^*, t^*) = \arg \min_{E(i,t) \in C(\tau)} t, \quad \text{where } C(\tau) = \{E(i,t) \mid \Delta_{E(i,t)}(\tau) = 1\}. \quad (2)$$

In this study, we will use this principle to identify the corresponding root cause of system failure. However, identifying the root cause is not a simple process, so we focus on analyzing, summarizing, and validating these root causes of failures.

## 3 THE **AGENTFAIL** DATASET

We collect systems execution data from two representative agentic AI development platforms, Dify and Coze, which provide the capabilities of visual workflow composition and tool integration. The dataset **AgentFail** contains 307 failure logs from ten platform-orchestrated agentic systems in total, with five from each platform. Each instance in the dataset includes four elements: (1) Query, a query obtained from the real test case; (2) Failure log: the full conversational trace of a system failing to complete the task. (3) System workflow and configuration: including the node orchestration structure and the information of each node, like agent's name, agent's prompt, code, tool configuration. (4) Annotations: the taxonomy labels and explanation of why the failure took place.

### 3.1 SYSTEM SELECTION

The agentic systems span a variety of task categories, including software development, information insight, task planning, and question & answering. Both platform-provided templates and user-defined systems we select from the platforms are included to ensure coverage of typical cases. To ensure diversity, we consider not only the task types but also the structural design of systems, covering five common categories: serial, parallel, branching, looping, and hybrid structures, whose details can be found in the Appendix A.3. This choice enables our analysis to cover diversified failure modes that are specific to both task semantics and orchestration patterns.

## 3.2 FAILURE LOG COLLECTION

Failure logs analyzed in this study are obtained from two sources: (1) open-source community contributions, where users had publicly shared. The example can be found in the Appendix A.6. From these reports, we extract both the user input and the corresponding execution results to construct failure cases; and (2) our own controlled runs, in which we execute public datasets (e.g., HumanEval Chen et al. (2021), TravelPlanner Xie et al. (2024)) or hand-crafted datasets based on the task, 1,387 test data in total, on corresponding agentic system and systematically record the traces. For hand-crafted datasets (denoted as Hand-crafted-platform name-Task name), we carefully design the construction criteria to ensure the representativeness and diversity of the inputs, which can be found in the Appendix A.5. For each run, success or failure is determined by the criteria which is mentioned in Section 2.2. Finally, we collect 307 failure logs in total and these combined sources ensure that our dataset captured both naturally occurring failures and reproducible benchmark failures. The detailed stats of **AgentFail** are listed in Table 1.

Table 1: Dataset Information. For test data sources with a large scale, we randomly select a subset of 100 samples to form the test set. We use the format `Hand-crafted-Platform Name-Task Name` to name the test data created by ourselves. The failure logs come from our run or community.

| Platform | Task | Structure | Test Data Source | Test Data size | Failure Log (Run + Community) |
|---|---|---|---|---|---|
| Dify | Code Generation | Looping | HumanEval | 163 | 23 (21+2) |
| | Program Repair | Looping | SWE-bench | 100 | 62 (62+0) |
| | Product QA Assistant | Branching | Hand-crafted-Dify-QA | 100 | 18 (17+1) |
| | Travel Assistant | Parallel | TravelPlanner | 180 | 32 (31+1) |
| | Deep Research | Hybrid | ninja-x-deepreasearch | 182 | 27 (26+1) |
| Coze | Product QA Assistant | Hybrid | Hand-crafted-Coze-QA | 100 | 37 (34+3) |
| | Travel Assistant | Parallel | TravelPlanner | 180 | 31 (28+3) |
| | Market Research Assistant | Serial | Hand-crafted-Dify-Market | 100 | 30 (29+1) |
| | Deep Research | Serial | ninja-x-deepreasearch | 182 | 27 (25+2) |
| | Industry Analysis | Parallel | Hand-crafted-Dify-Industry | 100 | 20 (20+0) |
| **Sum** | | | | **1,387** | **307 (293+14)** |

## 3.3 FAILURE ROOT CAUSE ANNOTATION

To systematically identify the root causes of failures, we adopt Grounded Theory (GT) annotation method Glaser & Strauss (1968), which is a qualitative research method that directly constructs theories from empirical data, to identify root cause patterns.

**Independent Annotation**. Three annotators with expertise in agentic systems independently examined system execution traces, where each annotator manually identifies the decisive errors and corresponding root cause based on the expert knowledge and understanding of failure logs. Additionally, each expert is instructed to separate their annotations into two categories: (i) cases in which they were fully confident about the correctness of the identified failure, and (ii) cases where they had any degree of uncertainty. The detailed information of this process is in Appendix A.7.

**Consensus Building**. Then, annotators focus specifically on the cases marked as uncertain in the first step. These instances are jointly reviewed, and through collaborative discussion, agreement is reached on the final labels.

**Cross Validation**. Finally, we adopt a cross-validation procedure. Each expert reviews the annotations made by others to assess the consistency of labeling standards. If any discrepancies are identified, the annotators engage in further discussion and, when necessary, re-annotate the data together until consensus is achieved. By incorporating multiple perspectives and enforcing agreement among annotators, this process enhances the reliability of the final annotations.

To assess the accuracy of the annotations, we calculate inter-annotator agreement using Cohen's $\kappa$, which changes from the initial 0.85 to the final 1.0. This indicates substantial agreement and validates that our GT-inspired procedure yielded reliable and reproducible attributions.

## 3.4 RELIABILITY OF ANNOTATION

To further validate the correctness of our taxonomy, we conduct targeted repair experiments based on the failure localization definition (Section 2.2.2) and counterfactual reasoning. The rationale is

that if applying the repair method for a specific failure root cause $\mathcal{D}$ converts the case from failure to success, the original labeling is correct. Our expert team design repair plans for these failure root causes grounded in their domain expertise and understanding of the system. Further details are provided in the Appendix A.10. We sample top half of failures based on occurrence frequency for verification, which can ensure the rationality of the experiment while reducing costs.

Figure 1 presents the confusion matrix of repair rates: rows correspond to the annotated failure labels, and columns correspond to the actual repairs applied. As shown in the figure, the diagonal values are consistently the highest (e.g., 90.1% for D1 (response formatting error), 95.6% for D2 (response content deviation), 96.3% for D5 (language or encoding defect)), indicating that repairs aligned with our annotations are the most effective. This result provides strong evidence for the accuracy and reliability of our annotations. We also observe off-diagonal effects which cannot be overlooked. For example, repairing D3 (knowledge or reasoning limitation) sometimes solves the failures labeled with D1 (response formatting error). This suggests that certain failure types share underlying dependencies or that one fix can alleviate multiple error pathways. Nevertheless, the clear dominance of diagonal entries demonstrates that our annotation captures the failure root causes in most cases, thereby validating the reliability of our taxonomy.

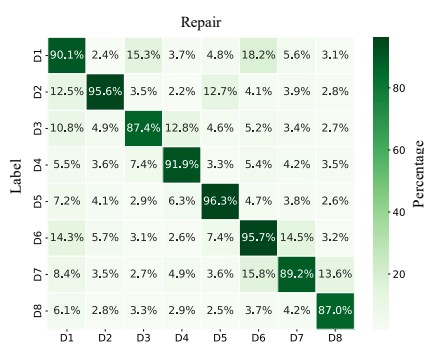

Figure 1: Confusion matrix of repair rates. Rows correspond to annotated failure root cause labels, and columns denote the applied repair. Higher diagonal values indicate that repairs aligned with annotations are most effective.

## 4 FAILURE ROOT CAUSE TAXONOMY

### 4.1 TAXONOMY

To systematically analyze the failure cases observed in platform-orchestrated agentic systems, we construct a three-level failure root cause taxonomy, as shown in Figure 2. This taxonomy is derived from iterative annotation of system execution traces and reflects the unique challenges posed by agent design, workflow orchestration, and platform environments. It enables fine-grained identification of failures to specific components, thereby facilitating diagnosis and repair. Agent-level failures capture failures that occur within a single agent, primarily due to limitations of the underlying language model or its interaction with local resources; Workflow-level failures arise from coordination or communication among multiple agents, often linked to workflow orchestration structures; Platform-level failures are attributable to the underlying platform or runtime environment.

### 4.2 ROOT CAUSE DISTRIBUTION

To better understand the characteristics of the failures, we conduct a statistical analysis of the failure root causes from several perspectives.

**Overall Distribution**. Firstly, we count the occurrences of each root cause, as shown in Figure 3a. Overall, we observe that agent-level failures dominate the dataset. Particularly, knowledge and reasoning limitations (F1.4) and poor prompt design (F1.5) are the most frequent categories, with more than fifty instances each. Other frequent categories include response format error (F1.2) and response content deviations (F1.3), reflecting the central role of LLM response quality in agentic systems. In contrast, workflow-level failures occur less frequently but still present important bottlenecks. Among these, missing input validation (F2.1) and unreasonable node dependency (F2.2) appear most often, indicating that lack of verification and orchestration design are common sources of error. Categories such as loops and deadlocks (2.3) or improper task decomposition (F2.5) are comparatively rare, yet they represent severe structural flaws when they do occur. Finally, platform failures account for a smaller proportion of failures. Both network and resource fluctuations (F3.1) and service unavailability (F3.2) are observed, though less frequently than agent- and workflow-level failures.

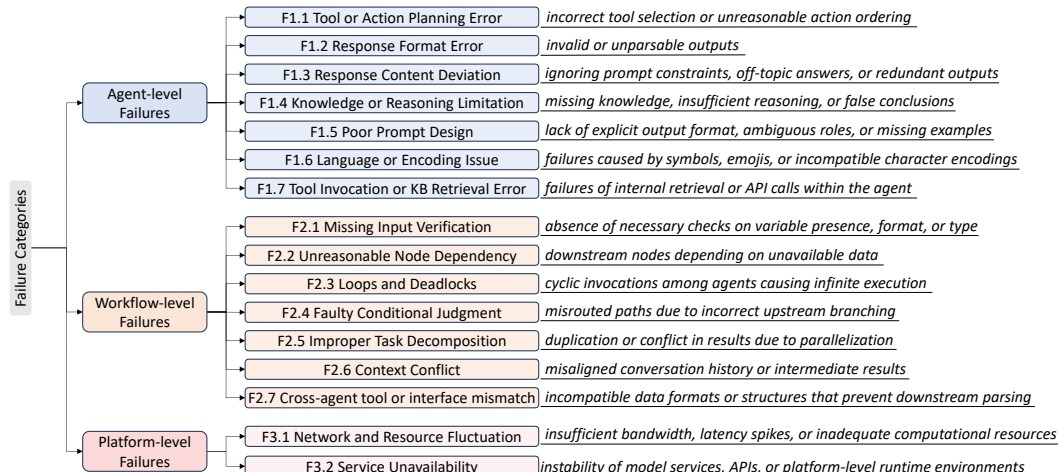

Figure 2: Failure Root Cause Taxonomy.

**Root Cause Distribution Across Platforms**. Secondly, we investigate differences in root cause distributions across platforms, as shown in Figure 3a. Among them, the darker three colors correspond to Dify, while the lighter three colors represent Coze. Overall, the two platforms exhibit broadly similar root cause distributions, with both dominated by agent-level issues (Types F1.x). However, a notable divergence appears in Type F2.2 (unreasonable dependencies), which occurs more frequently on Coze than on Dify. This discrepancy is largely attributable to two very long serial workflows on the Coze platform, each connecting multiple agents (market research assistant and deep search), whose complex dependency chains increase the likelihood of such failures. Apart from this case, the proportions of other workflow- and platform-level failures remain largely consistent across platforms, suggesting that the reliability challenges are platform-independent in most respects, with only certain workflow design mechanisms contributing to platform-specific variations.

**Root Cause Distribution Across Tasks**. Finally, we observe the difference in root cause distribution from the perspective of system tasks. For clarity, we categorize the ten systems into four groups: software development (code repair, code generation), task planning (travel assistant), Question & Answering, and information insight (deep research, industry analysis, marketing research). This categorization allows us to examine whether different application domains and system objectives are associated with distinctive failure patterns. The radar chart 3b reveals that different tasks exhibit distinct failure root cause modes. For software development, the most frequent failures arise from knowledge or reasoning limitations and prompt design issues, reflecting the heavy reliance on LLMs to generate and validate code. Question answering tasks, in contrast, are more prone to response format errors and tool invocation/retrieval errors, since these tasks often require precise data formatting and tool usage. Task planning tasks show a higher prevalence of language or encoding problems and unreasonable node dependencies, highlighting their sensitivity to control flow and

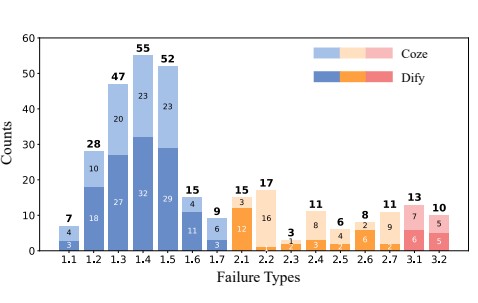

(a) Statistics of failure root cause per category.

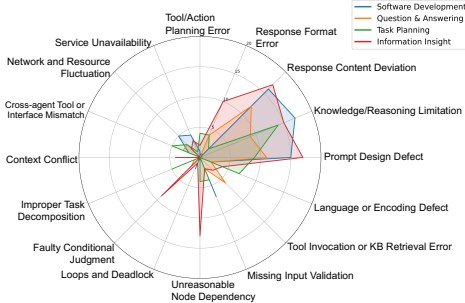

(b) Statistics of failure root cause per category across tasks.

Figure 3: Failure root cause distribution analysis from two perspectives: (a) overall distribution & across platforms and (b) across tasks.

Table 2: Impact of different failure root causes on system execution results. The values represent the average failure rate of agentic systems by injecting each failure root cause.

| Failure Root Cause | Execution Termination | Suboptimal Quality | Sum |
|---|---|---|---|
| Agent-level Failures | | | |
| F1.1 Tool or action planning error | 32.5% | 47.2% | 79.7% |
| F1.2 Response formatting error | 78.3% | 11.2% | 89.5% |
| F1.3 Response content deviation | 14.6% | 68.1% | 82.7% |
| F1.4 Knowledge or reasoning limitation | 2.1% | 72.5% | 74.6% |
| F1.5 Prompt design defect | 5.3% | 84.8% | 90.1% |
| F1.6 Language or encoding defect | 66.7% | 14.9% | 81.6% |
| F1.7 Tool invocation or KB retrieval error | 72.4% | 12.5% | 84.9% |
| Workflow-level Failures | | | |
| F2.1 Missing input validation | 51.2% | 21.8% | 73.0% |
| F2.2 Unreasonable node dependency | 67.9% | 9.4% | 77.3% |
| F2.3 Loops and deadlock | 65.4% | 6.7% | 72.1% |
| F2.4 Faulty conditional judgment | 41.8% | 29.6% | 71.4% |
| F2.5 Improper task decomposition | 18.2% | 50.5% | 68.7% |
| F2.6 Context conflict | 11.7% | 55.9% | 67.6% |
| F2.7 Cross-agent tool or interface mismatch | 76.2% | 10.3% | 86.5% |
| Platform-level Failures | | | |
| F3.1 Network and resource fluctuation | 84.9% | 4.7% | 89.6% |
| F3.2 Service unavailability | 88.1% | 3.2% | 90.3% |

dependency management. Meanwhile, information insight tasks, which typically involve multi-step reasoning and synthesis, are dominated by response content deviations and reasoning limitations, suggesting that subtle semantic errors accumulate throughout long reasoning chains.

## 4.3 IMPACT OF FAILURES

We investigate the extent to which individual failure root cause affect the overall performance of these agentic systems. We systematically inject one failure at a time according to the taxonomy introduced in Section 4.1. Each injected failure corresponds to a controlled modification, such as altering prompts, introducing incorrect conditional logic, or disabling input validation, while keeping all other components unchanged. By comparing the failure rates before and after failure injection, we quantify the performance degradation attributable to each failure root cause. This design allows us to identify the causal impact of individual failure root cause on workflow performance.

Since task failures may occur in different ways, and the potential impacts caused by their subsequent propagation are difficult to quantify, we categorize the outcomes of failures into two types and adopt the task failure rate as an objective metric to evaluate their impact, as described in Section 2.2.1. 1) **Execution Failure**: The execution is interrupted and cannot be completed; 2) **Suboptimal Quality**: The execution completes, but the quality of results does not meet expectations.

We count the number of occurrences of each failure root cause type in the two result categories to reflect the actual impact of the root cause on the system, as shown in Table 2. At the agent level, failure root causes like knowledge or reasoning limitations (F1.4) and prompt design defects (F1.5) tend to yield suboptimal outputs rather than outright failures, since the system can still execute the workflow but produces incomplete or low-quality answers. In contrast, response formatting (F1.2), language encoding errors (F1.6), or tool invocation failures (F1.7) directly disrupt the communication between components, often leading to execution termination. For workflow level, failure root causes such as loops and deadlocks (F2.3) or cross-agent interface mismatches (F2.7) often break the execution entirely, leading to very high termination rates. In contrast, improper task decomposition (F2.5) or context conflict (F2.6) typically allow the system to finish execution but impair the coherence and quality of intermediate results, resulting in suboptimal outputs. These differences suggest that structural problems in execution logic tend to cause termination, whereas coordination problems primarily degrade solution quality. At platform level, both failure root causes lead to high rates of execution failure. This indicates that once platform-level problems occur, they are more destructive compared to other failures, often leading to task termination rather than suboptimal quality.

Overall, the results reveal varying severity across failure categories: workflow- and platform-level failures often cause execution termination, while agent-level failures more often yield suboptimal quality. This distinction underscores the importance of tailoring failure root cause mitigation strategies according to the specific level at which the failure root cause arises.

## 5 AGENTFAIL BENCHMARK

Manual root cause identification is labor-intensive, motivating us to explore automated root cause identification with LLMs (`gpt-4o`, `LLaMA-3.1-70B`, `DeepSeek-R1`, `QWEN3-32B`, `GEMINI-2.5-PRO`,`CLAUDE-SONNET-4`), which covers open-source, closed-source model, reasoning model. Our goal is to examine whether our proposed taxonomy can assist LLMs in this process. We prompt LLMs to identify the root cause of failures under two settings: (1) without taxonomy, where LLMs rely solely on their own understanding of errors, and (2) with taxonomy, where the taxonomy and definitions are explicitly provided as guidance. By comparing the identification accuracy across these two settings, we can assess the extent to the taxonomy enhances the LLM's ability to diagnose the failure root cause.

Since failure logs are often long texts containing information from a large number of nodes, we adopt three different settings to evaluate their ability of identifying root causes, following the previous work Zhang et al. (2025d): **(1) All-at-once:** The LLM is given the query together with the full failure log and tasked with identifying the root cause of failure. **(2) Step-by-step:** Given a query and a step-by-step failure log, the LLM inspects each segment for the root cause. If identified, the process stops and returns it; otherwise, the next segment is examined. **(3) Binary search:** This method adopts a divide-and-conquer strategy: given the full log, the LLM judges whether the root cause lies in the first or second half, then recursively inspects the chosen half until the cause is identified. The three algorithms are detailed in Appendix A.8.

Table 3: Performance comparison on **AgentFail**.

| Models | Without Taxonomy | | | With Taxonomy | | |
|---|---|---|---|---|---|---|
| | All-at-once | Step-by-step | Binary Search | All-at-once | Step-by-step | Binary Search |
| `gpt-4o` | 9.6% | 11.5% | 9.8% | 27.4% | 31.2% | 28.3% |
| `GEMINI-2.5-PRO` | 9.8% | 11.0% | 10.2% | 24.1% | 25.0% | 24.4% |
| `CLAUDE-SONNET-4` | 11.7% | 12.0% | 11.7% | **30.2%** | 31.4% | 30.7% |
| `LLaMA-3.1` | 8.3% | 8.9% | 8.4% | 24.6% | 27.7% | 25.6% |
| `QWEN-32B` | 8.3% | 8.5% | 8.3% | 24.4% | 25.0% | 24.8% |
| `DeepSeek-R1` | 12.1% | 13.0% | 12.4% | 30.0% | **33.6%** | **31.4%** |

We show that providing the taxonomy itself facilitates root cause identification. Without the taxonomy, LLMs achieve only around 8.3%–13.0% accuracy in root cause identification. Incorporating the taxonomy substantially boosts performance to about 24.1%–33.6%, representing an absolute improvement of roughly 15–20 percentage points across models. This is because the taxonomy supplies LLMs with a structured guide to understand failures more reliably. We also observe that reasoning model like DeepSeek-R1 shows a slight edge, thanks to their ability to gradually break down problems. Despite this improvement, the peak accuracy remains at 33.6%, highlighting the inherent difficulty and challenging nature of automated root cause identification. The key challenge is that failure logs often contain long and complex contexts, where errors propagate through multiple nodes of the workflow. While LLMs may correctly flag errors at later stages, they struggle to trace back to the triggering node and the fundamental cause.

## 6 DISCUSSION

In light of the above experimental findings, we summarize the following actionable guidelines to help improve the reliability of platform-orchestrated agentic systems.

**Clear role specification and modular prompt design** can mitigate planning errors and response misalignments, two of the most frequent failure root cause categories we observed (related with F1.3 Response content deviation, F1.5 Prompt design defect).

**Explicit input and output validation** should be incorporated into nodes to prevent cascading errors from malformed data (related with F1.2 Response formatting error, F2.1 Missing input validation).

**Comprehensive checks or fallback mechanisms**, such as secondary validation agents or alternative tool paths, help solve the problems at the local level before they propagate across the workflow (related with F2.4 faculty conditional judgment). Sometimes, such comprehensive checks may introduce additional overhead, which requires a balance between robustness and efficiency.

**Progressive workflow design**, which starts with simple serial or parallel flows and gradually introduces complex patterns. It can help build more robust systems (related with F2.2 unreasonable node dependency, F2.3 loops and deadlock).

# 7 RELATED WORK

## 7.1 LLM MULTI-AGENT SYSTEMS

Contemporary LLM-based multi-agent systems can be broadly categorized by their degree of automation: (i) Hand-crafted systems, where the entire configuration (e.g., backbone LLMs, prompting strategies, and communication protocols) is explicitly specified, as exemplified by AutoGen Wu et al. (2023b), Camel Li et al. (2023), and ChatDev Qian et al. (2024). (ii) Partially-automated systems, which automate specific components: for instance, AutoAgents Chen et al. (2024), LLMSelector Chen et al. (2025), and MasRouter Yue et al. (2025) assign agent roles automatically; DsPy Khattab et al. (2024) and TextGrad Yuksekgonul et al. (2025) optimize prompt design; GPTSwarm Zhuge et al. and G-Designer Zhang et al. (2025b) adaptively construct inter-agent topologies. (iii) Fully automated systems, where all modules are automatically designed and evolved Hu et al. (2025); Zhang et al. (2025c); Wu et al. (2025); Nie et al. (2025). The systems studied in this paper, which are built on low-code platforms primarily fall into the first category of hand-crafted systems. However, they represent a distinct subclass defined by visual, template-driven configuration rather than pure code, which introduces unique challenges in design, debugging, and repair.

## 7.2 FAILURE IDENTIFICATION FOR AGENTIC SYSTEMS

With the increasing complexity of multi-agent systems (characterized by multiple agents Wang et al. (2025), tool integration Shen et al. (2024), and communication protocols Marro et al. (2024)), widespread errors and structural vulnerabilities have become urgent issues. Zhang et al. (2025d) proposed the Who&When dataset, which contains failure logs from LLM MAS, annotated with the responsible agent and step, and evaluated automatic attribution methods. Ge et al. (2025) proposed a spectrum-based approach to estimate each agent's "suspiciousness" via trajectory replay and spectral analysis. Cemri et al. (2025) proposed the MAST taxonomy, showing that failures in LLM MAS often arise from specification flaws, inter-agent misalignment, and weak verification. Zhang et al. (2025a) developed AgenTracer and the TracerTraj dataset to enable fine-grained error diagnosis through counterfactual replay and reinforcement learning. However, existing methods mainly focus on locating the step where a failure occurs but fall short of diagnosing the failure root cause, such as a bad prompt, incorrect task decomposition, or a logical deadlock. Our work addresses this gap by arguing that a clear taxonomy of root causes is prerequisite to moving from superficial failure localization to a deeper understanding of system failures.

# 8 CONCLUSION

In this work, we present **AgentFail**, a dataset of 307 annotated failure logs from platform-orchestrated agentic systems, and ensure the reliability of root cause annotation through counterfactual repair experiments. Building on this dataset, we propose a fine-grained taxonomy of agent-, workflow-, and platform-level failure root causes, and show that while agent-level failures dominate, workflow dependencies also introduce substantial risks, particularly in long serial structures. Furthermore, we benchmark the proposed taxonomy by integrating it into automatic root cause identification with LLMs, demonstrating that it can promote automated diagnosis of the root causes. Overall, our study not only advances the empirical understanding of failure root causes in agentic systems, but also provides actionable insights and benchmarks that support the development of more robust and reliable agentic systems.

ETHICS STATEMENT

This work does not involve human subjects, personal data, or sensitive information. All failure logs used in our dataset were generated from controlled executions of platform-orchestrated agentic systems or open-source platforms. The dataset contains only synthetic task inputs, system outputs, and annotations produced by domain experts, ensuring that no personally identifiable or confidential information is included.

Overall, this work aims to promote reliability in the development of agentic systems by offering resources for diagnosing and understanding their failure modes, thereby contributing to safer and more reliable deployment.

REPRODUCIBILITY STATEMENT

We will release all the dataset and code used in our work. These resources enable other researchers to replicate our results and extend our study.

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

## A APPENDIX

### A.1 THE USE OF LARGE LANGUAGE MODELS (LLMS)

In preparing this paper, we used LLMs solely as a writing assist tool, specifically for grammar checking and minor language refinement. The models were not involved in research ideation, experiment design, analysis, or substantive writing. The authors take full responsibility for all content.

### A.2 BACKGROUND

The detailed description of the node element are as follows:

1) Start: The Start node is a preset node that is required for each workflow application, which provides the necessary initial information for subsequent workflow nodes and the normal flow of the application, such as the content entered by the application user and the uploaded files.

2) End: Define the final output of a workflow end. Each workflow requires at least one end node after full execution to output the final result of full execution. The end node is the process termination node, and no other nodes can be added later. If a conditional fork occurs in the process, multiple end nodes need to be defined. The end node needs to declare one or more output variables, which can refer to the output variables of any upstream node.

3) Answer: You can freely define the format of your reply in the text editor, including customizing a fixed piece of text, using the output variables in the prelude as the reply content, or combining custom text with variables. You can add nodes at any time to stream content to the conversation reply, support WYSIWYG configuration mode, and support graphic mixing, such as: Output the LLM node reply, Output generates images, Output plain text.

4) LLM: The ability to invoke large language models to process information (natural language, uploaded files, or images) entered by users in the Start node to provide effective response information.

5) Knowledge retrieval: Retrieve text content related to user issues from the knowledge base and use it as the context of downstream LLM nodes.

6) Question Classifier: By defining classification descriptions, problem classifiers can use LLMs to infer matching classifications based on user input and output classification results, providing more accurate information to downstream nodes.

7) IF/ELSE: Split the workflow process into multiple branches based on If/else/elif criteria.

8) Code: Code nodes support running Python / NodeJS code to perform data transformations in workflows. It streamlines your workflow and is suitable for Arithmetic, JSON transform, text processing, and more. This node greatly enhances developer flexibility, allowing them to embed custom Python or Javascript scripts in their workflows and manipulate variables in ways that preset nodes cannot. By configuring the options, you can specify the required input and output variables and write the appropriate execution code.

9) Template: Allows flexibility for data transformation, text processing, and more with the help of Jinja2's Python templating language.

10) Variable Aggregator: Aggregate the variables of multiple branches into a single variable to achieve unified configuration of downstream nodes. The variable aggregation node (original variable assignment node) is a key node in the workflow, responsible for integrating the output results of different branches, ensuring that no matter which branch is executed, its results can be referenced and accessed through a unified variable. This is useful in the case of multi-branching, which allows you to map variables with the same effect in different branches to one output variable, avoiding duplicate definitions by downstream nodes.

11) Parameter Extractor: Leverage LLMs to reason from natural language and extract structured parameters for back-up tool calls or HTTP requests. The Dify workflow provides a rich selection of tools, most of which have structured parameters as inputs, and the parameter extractor can convert the user's natural language into parameters that the tool can recognize for easy tool calling. Some nodes in the workflow have specific data format input requirements, such as the input requirements of iteration nodes in array format, and the parameter extractor can easily realize the conversion of structured parameters.

12) Iteration: Performing the same operation steps on elements in the array in turn until all results are output, which can be understood as a task batch processor. Iteration nodes are usually used with array variables. For example, in the long-text translation iteration node, if you input everything into the LLM node, you may reach the single conversation limit. Upstream nodes can first split long texts into multiple fragments and perform batch translation of each fragment with iteration nodes to avoid reaching the message limit of LLM single conversation.

13) HTTP Request: Allows server requests to be sent over the HTTP protocol, which is suitable for obtaining external data, webhooks, generating images, downloading files, etc. It allows you to send customized HTTP requests to specified network addresses, enabling interconnection with various external services. This node supports common HTTP request methods, such as GET, POST and so on.

14) Tools: Tools nodes can provide powerful third-party capability support for workflows in three types: Built-in tools, Dify first-party tools, may require authorization before using the tool. Custom tools, tools that are imported or configured via the OpenAPI/Swagger standard format. If the built-in tools don't meet your needs, you can create a custom tool in the Dify menu navigation –Tools. Workflows, where you can orchestrate a more complex workflow and publish it as a tool. For more information, refer to the tool configuration instructions.

15) Variavle Assigner: The Variable Assignment node is used to assign variables to writable variables, and the following writable variables are supported: Converation variables and cyclic variables.

16) Loop: Loop nodes are used to execute recurring tasks that depend on the results of the previous round until the exit conditions are met or the maximum number of loops is reached.

## A.3 DIFFERENT STRUCTURES

Agentic workflows can be organized under different architectural paradigms, each reflecting a distinct way of coordinating agents and tools:

1. Serial: Tasks are executed in a sequential manner, where the output of one node serves directly as the input to the next. This design is simple and interpretable but suffers from error propagation, as mistakes in early steps cascade downstream.

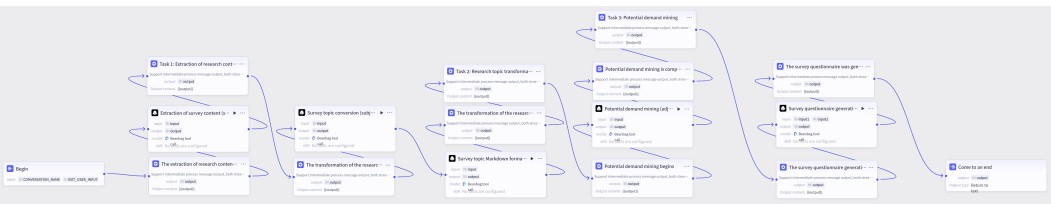

Figure 4: Example of serial structure.

2. Parallel: Multiple nodes process subtasks simultaneously, and their outputs are later aggregated. This improves efficiency and robustness but can lead to synchronization issues or inconsistencies in the merged results.

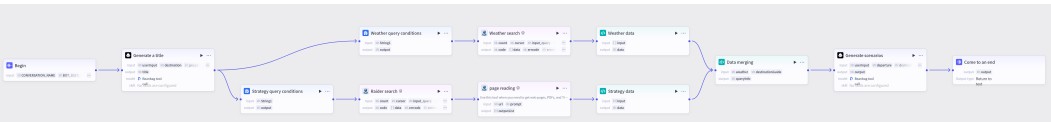

Figure 5: Example of parallel structure.

3. Branching: The workflow splits into different paths based on conditions or task types. While this design enables flexibility and specialization, it may suffer from dependency mismatches or logic errors across branches.

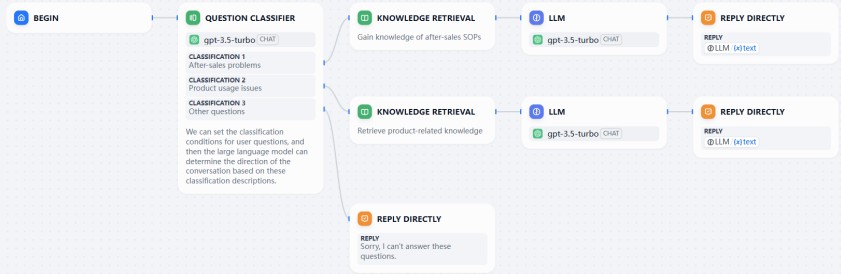

Figure 6: Example of branching structure.

4. Looping: Certain steps are repeated until a stopping condition is satisfied, enabling refinement and self-correction. However, this design is prone to infinite loops or redundant computation.

5. Hybrid: Real-world workflows often combine multiple patterns, such as sequential backbones with parallel or looping substructures. Hybrid designs increase flexibility and expressiveness but also complicate debugging and error attribution.

## A.4 MULTI-LLM JUDGE

In evaluating system executions, we draw inspiration from the LLM-as-a-Judge paradigm, where large language models are employed to assess the correctness of outputs in the absence of deterministic ground truth. To mitigate potential bias from relying on a single model, we further adopted a multi-LLM judging strategy, using several independent models to provide parallel assessments.

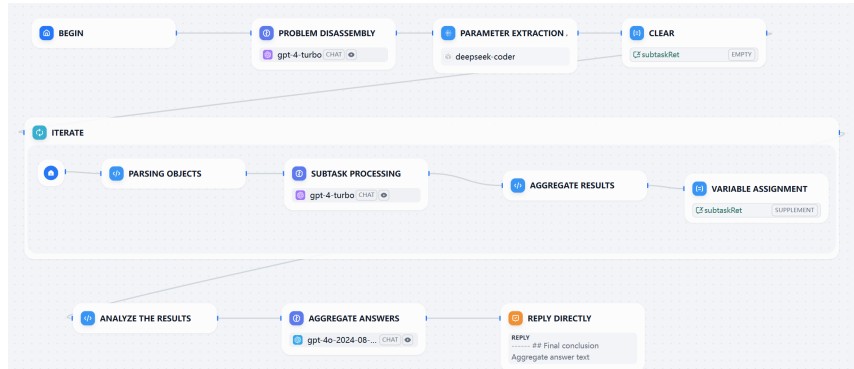

Figure 7: Example of looping structure.

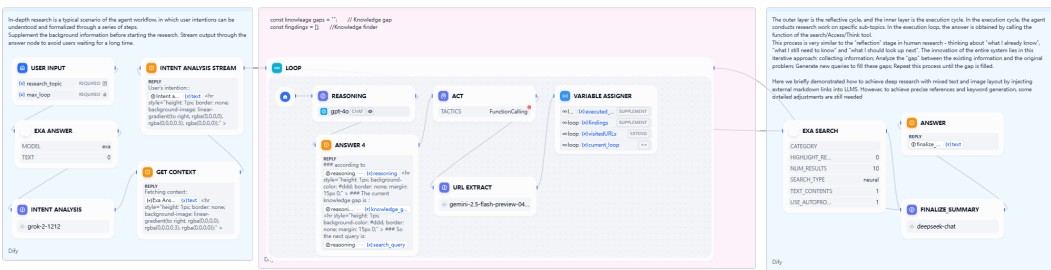

Figure 8: Example of hybrid structure.

The final decision was derived through majority voting across models. Specifically, we employed GPT-4o, Claude 3.7, DeepSeek, and Gemini, selected for their strong reasoning and complementary capabilities. This ensemble of models ensures greater robustness and helps mitigate systematic biases inherent to any individual model. Example of the prompt we design is as following:

---

**Prompt**

You are a professional industry research evaluator. Please analyze and score the quality of the assistant's industry analysis output based on the following dimensions:
Relevance: Does the analysis stay focused on the given industry and research question?
Depth of Analysis: Does it go beyond surface-level description to include trends, drivers, risks, and opportunities?
Use of Evidence: Does the analysis reference data, examples, or case studies to support its claims?
Actionability: Does it provide insights that can inform business or strategic decisions?
Clarity & Structure: Is the analysis well-structured, logically coherent, and easy to understand?
Completeness: Does the analysis cover all key perspectives (market size, competition, user behavior, risks, opportunities)?
For each dimension, provide:
A score from 1 to 5 (5 = best)
A one-sentence overall evaluation of the research result
Output format example:

Industry: {question}
Assistant's Industry Analysis: {answer}

{{
  "Relevance": 4,
  "Depth of Analysis": 5,
  "Use of Evidence": 3,
  "Actionability":  4,
  "Clarity & Structure": 5,
  "Completeness": 4,
  "Score": 25,
  "Comment": "A strong analysis with actionable insights, though it could be improved with more supporting evidence."
}}

---

Figure 9: Example of prompt

## A.5 TEST DATA CONSTRUCTION

To ensure the diversity and representativeness of the test input set, we systematically construct test cases based on the following four dimensions:

1) **Functional path coverage.** Based on the official documentation of the workflow and the node topology, we identified all the key functional paths. The test input is designed to trigger different conditional branches (for example, in the travel planning assistant, single-destination, multi-destination, and requests containing invalid destinations are tested respectively) to ensure coverage of the main, edge, and error handling logic of the workflow.

2) **Input mode and complexity.** We took into account multiple modalities and complexities of the input information, including:

Text complexity: From simple keyword queries to complex long texts containing multi-round dialogue contexts.

Degree of structuring: Test unstructured natural language input against partially structured (such as lists) or fully structured (such as those conforming to a specific JSON Schema) input.

3) **Data Augment.** For agentic systems that contain knowledge bases (such as question-answering assistants), we conduct systematic semantic transformation on the original questions in the knowledge base to generate test inputs that are semantically equivalent but diverse in expression. This method not only expands the scale of the test set, but more importantly, assesses the system's ability to understand the language diversity in users' real queries.

The conversion strategies we adopt include:

1) Use pre-trained language models (such as Sentence-BERT) to retrieve synonyms and make replacements, while changing the active/passive voice or word order of the Sentence (for example, convert "How long is the battery life of this mobile phone?" to "How long can the battery of this mobile phone last? ")

2) To efficiently generate large-scale and high-quality semantic conversion test cases, we have adopted a LLM as the semantic engine. The specific process is as follows:

- Seed input: Use the original questions in the knowledge base as seeds.

- Instruction-controlled generation: Structured prompts (e.g., "Rephrase this question in three different styles," "Simulate novice vs. expert phrasing," "Embed the question in a dialogue context") guided the LLM to produce varied but semantically equivalent queries.

- Automated filtering and deduplication: Generated candidates were clustered and filtered by embedding similarity to ensure diversity and quality.

## A.6 EXAMPLE OF COMMUNITY ISSUE REPORT

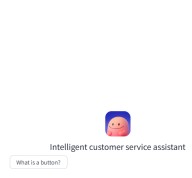

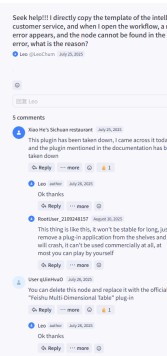

## A.7 ANNOTATION

Figure 10a shows the proportion of uncertain annotations to total annotations during the first round. Figure 10b shows the initial disagreement rates between annotators (note that we make sure to reach a consensus through a careful discussion and voting process afterwards).

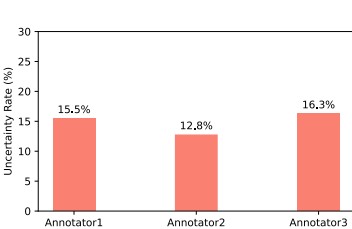
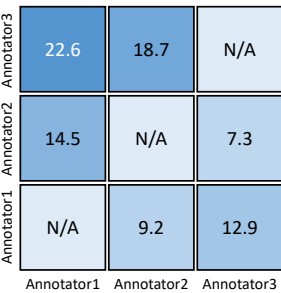

(a) Uncertain annotation percentages.      (b) Disagreement rates in voting.

Figure 10: Statistical analysis of the annotation process: (a) The proportion of uncertain annotations to total annotations during the independent annotation. (b) Initial disagreement rates among annotators (with final consensus achieved through subsequent discussion and voting).

## A.8 ALGORITHM DETAILS

We provide more details on the Step-by-Step and Binary Search root cause identification methods. We use $Q$ to represent the query provided to the system. $L = \{l_1, l_2, \ldots, l_n\}$ denotes failure log consisting of $n$ entries where each entry $l_i$ specifies the input and output of one node at time step $i$ in the system.

---

**Algorithm 1** Step-by-Step

---

**Require:** Query $Q$, failure log $L = \{l_1, l_2, \ldots, l_n\}$
**Ensure:** Root cause $R^*$
1: **for** $i \in \{1, 2, \ldots, n\}$ **do**
2:      Provide $Q$ and $\{l_1, \ldots, l_i\}$ to LLM
3:      **if** LLM indicates root cause at step $i$ **then**
4:          **return** $R^*$
5:      **end if**
6: **end for**
7: No failure found

---

## A.9 PROMPTS

**Prompts of All-at-Once**

**Prompts of Step-by-step**

**Prompts of Binary Search**

## A.10 MANUAL OPTIMIZATION METHODS

- For response formatting errors, we enforced strict JSON output mode at the LLM node and appended a code-execution node for format validation, with automatic re-prompting upon failure.
- For response content deviation, we incorporated rule-based or embedding-based semantic similarity checks, followed by a correction prompt when the deviation exceeded a preset threshold.

**Prompt without taxonomy**

You are an AI assistant tasked with analyzing an agentic system built on platform conversation history when solving a real world problem. You should use the following information to identify the root cause of the failure.
The problem is: {problem}.
The system is : {system description}
Here's the conversation: {failure log}
Based on this conversation, please predict the following:
The root cause of the failure: {Root Cause}
The reason for your prediction: {Reason}

**Prompt with taxonomy**

You are an AI assistant tasked with analyzing an agentic system built on platform conversation history when solving a real world problem. Please refer to the taxonomy I provided.
You should choose one of them as the root cause of failure that you identify.
The problem is: {problem}.
The system is : {system description}
Here's the conversation: {failure log} and the taxonomy {taxonomy}
Based on this conversation and taxonomy, please predict the following:
The root cause of the failure: {Root Cause}
The reason for your prediction: {Reason}

**Prompt without taxonomy**

You are an AI assistant tasked with analyzing an agentic system built on platform conversation history when solving a real world problem. You should use the following information to identify the root cause of the failure.
The problem is: {problem}.
The system is : {system description}
Here's the part of conversation history: {failure log}
Based on this conversation, please predict the following:
The root cause of the failure: {Root Cause}
The reason for your prediction: {Reason}

**Prompt with taxonomy**

You are an AI assistant tasked with analyzing an agentic system built on platform conversation history when solving a real world problem. Please refer to the taxonomy I provided. You should use the following information to identify the root cause of the failure.
The problem is: {problem}.
The system is : {system description}
Here's the part of conversation history: {failure log} and the taxonomy {taxonomy}
Based on this conversation, please predict the following:
The root cause of the failure: {Root Cause}
The reason for your prediction: {Reason}

**Prompt without taxonomy**

You are an AI assistant tasked with analyzing an agentic system built on platform conversation history when solving a real world problem. You should use the following information to identify the root cause of the failure.
The problem is: {problem}.
Review the following conversation range {range description}: {sliced log}.
Based on your analysis, predict whether the error is more likely to be located in the upper or lower half of the segment. lower half is defined as the range lower half range and upper half is defined as the range upper half range. Please simply output either 'upper half' or 'lower half'. You should output the root cause until you think you find it.

**Prompt with taxonomy**

You are an AI assistant tasked with analyzing an agentic system built on platform conversation history when solving a real world problem. Please refer to the taxonomy I provided. You should use the following information to identify the root cause of the failure.
The problem is: {problem}.
Review the following conversation range {range description}: {sliced log}.
Based on your analysis, predict whether the error is more likely to be located in the upper or lower half of the segment. lower half is defined as the range lower half range and upper half is defined as the range upper half range. Please simply output either 'upper half' or 'lower half'. You should not output anything else. You should output the root cause until you think you find it.

---

**Algorithm 2** Binary Search

---

**Require:** Query $Q$, failure log $L = \{l_1, l_2, \ldots, l_n\}$
**Ensure:** Root cause $R^*$
1: Initialize $low \leftarrow 1$, $high \leftarrow n$
2: **while** $low < high$ **do**
3:     $mid \leftarrow \left\lfloor \frac{low+high}{2} \right\rfloor$
4:     Extract log segment $L' \leftarrow \{l_{low}, l_{low+1}, \ldots, l_{mid}\}$
5:     Provide $Q$ and $L'$ to LLM
6:     **if** LLM indicates root cause in $L'$ **then**
7:         $high \leftarrow mid$
8:     **else**
9:         $low \leftarrow mid + 1$
10:    **end if**
11: **end while**
12: $R^* \leftarrow low$
13: **return** $R^*$

---

- For LLM knowledge or reasoning limitation, we substituted the original model with a more powerful or domain-adapted LLM.

- For prompt design defect, we established a standardized prompt template including roles, input–output formats, boundary conditions, illustrative examples, and error cases.

- For language or encoding defect, we added explicit encoding normalization steps (e.g., UTF-8 enforcement) and inserted a preprocessing node to unify tokens, scripts, or mathematical symbols before model consumption.

- For missing input validation, we added explicit input checks to ensure required variables, types, and formats were satisfied.

- For unreasonable node dependency, we reconstruct the workflow design to adjust task allocation and reduce redundancy or conflicts.

- For network and resource fluctuations, we augmented LLM nodes with fallback branches that automatically degrade to backup models when failures are detected.

