# OpenReview forum: "Diagnosing Failure Root Causes in Platform-Orchestrated Agentic Systems: Dataset, Taxonomy, and Benchmark"
_ICLR.cc/2026/Conference — ICLR 2026 Conference Withdrawn Submission_

### Official Review · Reviewer_Y37G · 2025-10-31

**Soundness:** 2
**Presentation:** 2
**Contribution:** 1
**Rating:** 2
**Confidence:** 4

**Summary:**

This paper studies the failure mechanisms of platform-orchestrated agentic systems — systems built via low-code platforms such as Dify and Coze, where multiple LLM agents coordinate through structured workflows.
The authors propose AgentFail, a dataset of 307 failure logs collected from ten systems across two platforms. Each case is annotated with the root cause of failure (using a grounded-theory annotation process and verified through counterfactual repair).
They further introduce a three-level taxonomy (agent-level, workflow-level, and platform-level failures) and an LLM-based benchmark to automatically identify the root cause. Experiments show that providing the taxonomy improves identification accuracy from about 10% to 33%, but the task remains difficult.

**Strengths:**

1. Relevance and motivation: The paper studies an emerging and practical problem: why multi-agent LLM systems built via platforms fail. With tools like Dify and Coze getting popular, understanding their fragility is indeed necessary.

2. Paper writing
The paper is overall well-writing and easy to follow.

**Weaknesses:**

1. Limited originality
While the paper claims novelty, it heavily overlaps in spirit with Zhang et al., 2025d (“Which Agent Causes Task Failures and When?”) and Cemri et al., 2025 (“Why Do Multi-Agent LLM Systems Fail?”). The “root cause” concept here appears as an extension or recombination of those works, not a fundamentally new paradigm. This work seems like a stitched combination of prior benchmarks (Who&When + MAST) applied to Dify/Coze.

2. Conceptual overlap and presentation
Several parts (taxonomy diagrams, repair analysis) are nearly identical to those in previous works, differing only in labels or platforms. The conclusion "failure attribution is challenging" has already been pointed out by previous works. So I do not understand what is the real conclusion of this paper.

3. Dataset scale and diversity
Although the dataset has 307 failures, this is relatively small given ten systems and multiple task types. The number of distinct workflows per platform (five each) may not generalize well to other agentic frameworks (e.g., LangChain, HuggingGPT). The scope is narrow and may not support strong general claims about **"platform-orchestrated"** systems.

**Questions:**

1. The authors should clearly articulate how this work differs from Zhang et al., 2025d (“Which Agent Causes Task Failures and When?”) and Cemri et al., 2025 (“Why Do Multi-Agent LLM Systems Fail?”).

2. Taxonomy construction:
Was the taxonomy inductively derived or guided by existing taxonomies like MAST?

---

### Official Review · Reviewer_EPDZ · 2025-10-31

**Soundness:** 2
**Presentation:** 2
**Contribution:** 2
**Rating:** 2
**Confidence:** 5

**Summary:**

The paper assembles a dataset of 307 failure logs from ten multi-agent systems built on two low-code platforms, Dify and Coze. It proposes a three-level taxonomy of root causes spanning agent, workflow, and platform sources, and evaluates automatic root-cause identification by prompting off-the-shelf LLMs with or without the taxonomy.

**Strengths:**

1. Focus on platform-orchestrated systems is well motivated and reflects how many users build agents today.
2. The dataset spans several workflow topologies and task families, which is a reasonable starting point for analysis.

**Weaknesses:**

1. The core weakness of the paper is limited novelty relative to recent work. The contribution reads as an amalgam of prior studies. Like [1] and related attribution papers [2], it runs multi-agent systems and analyzes failure traces. Like [2], it introduces a categorization of failure patterns. It also echoes prior efforts in annotator analysis and uses off-the-shelf LLMs to diagnose logs. The annotation protocal is almost a duplicate of [1]. The high-level takeaway appears unchanged from those lines of work, and the new taxonomy feels largely isomorphic to existing ones with new labels rather than new insights.
2. The method defines the root cause as the earliest decisive error that flips failure to success when counterfactually corrected, which is convenient but can ignore latent upstream design flaws like missing validation that merely manifested later.

[1] Which agent causes task failures and when? On automated failure attribution of llm multi-agent systems

[2] Why Do Multi-Agent LLM Systems Fail?

**Questions:**

1. The benchmark shows only modest gains when the taxonomy is given to the model. What additional signal or supervision would be needed to reach practitioner-useful accuracy?
2. Can the taxonomy guide automatic repair, not just diagnosis? A short case study demonstrating end-to-end bug fixing would strengthen the practical contribution.

---

### Official Review · Reviewer_XsZm · 2025-11-01

**Soundness:** 2
**Presentation:** 3
**Contribution:** 1
**Rating:** 4
**Confidence:** 5

**Summary:**

This paper investigates how to identify and analyze failures in platform-orchestrated agentic systems—multi-agent setups built using low-code tools like Dify. The authors introduce AgentFail, a dataset of 307 annotated failure cases, and develop a taxonomy of root causes to guide both human and LLM-based diagnosis.

**Strengths:**

The paper introduces a well-structured taxonomy that categorizes different root causes of failures in agentic systems. By using counterfactual reasoning to validate annotations, the study ensures higher reliability and rigor in its dataset

**Weaknesses:**

A key weakness of this paper is that the AgentFail dataset and benchmark are not substantially different or larger than existing benchmarks on LLM-based agent failures. While it provides valuable annotations and taxonomy, the dataset’s modest size (307 logs) and similar structure to prior diagnostic benchmarks limit its novelty and scalability, making it less impactful for evaluating generalizable failure analysis across diverse agentic systems.

Another weakness is that it’s unclear whether introducing a separate classification task for root cause identification adds meaningful value beyond standard error attribution analyses.

**Questions:**

See weakness

---

### Official Review · Reviewer_pc4Z · 2025-11-07

**Soundness:** 2
**Presentation:** 2
**Contribution:** 3
**Rating:** 4
**Confidence:** 4

**Summary:**

This paper introduces a study to diagnose the failure root causes in multi-agent systems built on low-code platforms. The authors contribute AgentFail, a dataset of 307 annotated failure logs, and a new taxonomy that classifies failures at the agent, workflow, and platform levels. While this taxonomy significantly improves the ability of LLMs to automatically find the root cause, the task remains highly challenging, with the best-performing models only achieving 33.6% accuracy.

**Strengths:**

The paper’s primary strength is its systematic study of failure root causes in platform-orchestrated (low-code) agentic systems. It contributes the AgentFail dataset, featuring root-cause labels tied to execution traces that are uniquely validated using a “counterfactual repair” methodology. This work establishes a concrete benchmark for automated root cause identification in this specific context, quantifying its high difficulty (33.6% max accuracy) and providing a clear baseline for future research.

**Weaknesses:**

Although the authors outline a GT-inspired multi-annotator process, the taxonomy construction still reads relatively ad-hoc: the criteria for defining, merging, or splitting categories and concrete coding examples are insufficiently detailed. The core benchmark is methodologically shallow; the diagnostic “method” is simply prompting existing LLMs and lacks any novel algorithmic contribution or comparison against strong, non-LLM baselines (e.g., rule-based or supervised classifiers). This lack of depth, combined with a failure to clearly differentiate from related work on agent failure and inconsistent notation, makes the work feel more like a descriptive study than a complete technical paper.

**Questions:**

1. Your repair validation (Fig 1) shows off-diagonal effects—e.g., a repair for D3 (reasoning) also fixes D1 (formatting). Doesn’t this confound the experiment and undermine the core claim that your taxonomy categories are distinct and identifiable?

2. The 33.6% accuracy benchmark is difficult to interpret. To distinguish inherent task difficulty from the weakness of your prompting method, please provide two crucial baselines: (a) the performance of a simpler, non-LLM classifier (e.g., fine-tuned BERT or keyword-based), and (b) the human expert accuracy for this same task.

3. The novelty claim of “why” (root cause) vs. “where” (localization) appears weak, as other cited work (e.g., Cemri et al.) also proposes root-cause taxonomies. Please articulate the specific, novel contribution of your taxonomy itself, beyond its application to low-code platforms.

4. The paper’s credibility is weakened by inconsistent notation and numerous typos. For instance, Section 3.4 and Figure 1 introduce a new “D1, D2...” notation that clashes with the “F1.x” taxonomy used elsewhere. Please clarify this discrepancy and thoroughly proofread the manuscript.

---

### Note · Authors · 2025-11-19

I have read and agree with the venue's withdrawal policy on behalf of myself and my co-authors.